# High Expression of ENO1 and Low Levels of Circulating Anti-ENO1 Autoantibodies in Patients with Myelodysplastic Neoplasms and Acute Myeloid Leukaemia

**DOI:** 10.3390/cancers16050884

**Published:** 2024-02-22

**Authors:** Lisa F. Lincz, Danielle Z. Theron, Daniel L. Barry, Fiona E. Scorgie, Jonathan Sillar, Opelo Sefhore, Anoop K. Enjeti, Kathryn A. Skelding

**Affiliations:** 1Haematology Department, Calvary Mater Newcastle, Waratah, NSW 2298, Australia; fiona.scorgie@calvarymater.org.au (F.E.S.); jonathan.sillar@calvarymater.org.au (J.S.); opax24@yahoo.com (O.S.); anoop.enjeti@calvarymater.org.au (A.K.E.); 2University of Newcastle, University Drive, Callaghan, NSW 2308, Australia; danielle.theron@uon.edu.au (D.Z.T.); daniel.l.barry@uon.edu.au (D.L.B.); kathryn.skelding@newcastle.edu.au (K.A.S.); 3Hunter Medical Research Institute, Lookout Road, New Lambton, NSW 2305, Australia; 4New South Wales Health Pathology, John Hunter Hospital, Lookout Road, New Lambton, NSW 2305, Australia

**Keywords:** alpha-enolase, ENO1, autoantibodies, acute myeloid leukaemia, myelodysplastic neoplasms, bone marrow, biomarker

## Abstract

**Simple Summary:**

Acute myeloid leukaemia (AML) is the most common acute leukaemia affecting adults and has one of the poorest survival rates of all cancers. The disease is often pre-cursed by myelodysplastic neoplasms (MDS), a more benign class of diseases that can last many years but also carry a high risk of progression to AML. Diagnosis requires histological examination of a bone marrow biopsy, an invasive procedure that can be influenced by sampling variations and subjective enumeration. A circulating biomarker that could be measured as part of a routine peripheral blood test would provide a much quicker, safer, and reliable alternative. This study examines the potential use of α-enolase (ENO1) as a potential biomarker and shows that it is highly expressed in bone marrow biopsies from patients with both MDS and AML, and that patients with these diseases also have decreased circulating levels of autoantibodies to ENO1.

**Abstract:**

In solid tumours, high expression of the glycolytic enzyme, α-enolase (ENO1), predicts for poor patient overall survival (OS), and circulating autoantibodies to ENO1 correlate positively with diagnosis and negatively with advanced disease. Although ENO1 is one of the most highly expressed genes in acute myeloid leukaemia (AML), its potential role as a biomarker in AML or its precursor, myelodysplastic neoplasms (MDS), has not been investigated. A meta-analysis of nine AML online datasets (n = 1419 patients) revealed that high ENO1 expression predicts for poor OS (HR = 1.22, 95% CI: 1.10–1.34, *p* < 0.001). Additionally, when compared to AML in remission (n = 5), ENO1 protein detected by immunohistochemistry was significantly higher at diagnosis in bone marrow from both AML (n = 5, *p* < 0.01) and MDS patients (n = 12, *p* < 0.05), and did not correlate with percentage of blasts (*r* = 0.28, *p* = 0.21). AML patients (n = 34) had lower circulating levels of ENO1 autoantibodies detected by ELISA compared to 26 MDS and 18 controls (*p* = 0.003). However, there was no difference in OS between AML patients with high vs. low levels of anti-ENO1 autoantibodies (*p* = 0.77). BM immunostaining for ENO1 and patient monitoring of anti-ENO1 autoantibody levels may be useful biomarkers for MDS and AML.

## 1. Introduction

Acute myeloid leukaemia (AML) is the most common acute leukaemia affecting adults and has one of the poorest survival rates of all cancers with a median five year survival of just 24% [1]. The disease is often pre-cursed by myelodysplastic neoplasms (MDS), a heterogeneous group of clonal myeloid disorders characterised by cytopenia and dysplastic changes in bone marrow haematopoietic cells [2]. These diseases typically affect the elderly and carry a high risk of transformation to AML [3]. Both AML and MDS are underpinned by changes in the bone marrow microenvironment that drive disease progression at the expense of normal haematopoietic cell function [4]. The pathological overlap between the two diseases has recently been officially recognised by the updated 2022 International Consensus Classification (ICC) by creating new categories of MDS/AML [5]. While MDS, MDS/AML, and AML may share many genomic abnormalities, they remain differentiated by bone marrow myeloid blast count by both the ICC and World Health Organization classifications [5,6]. Assessment of such an arbitrary cut-off requires histological examination of a bone marrow biopsy and can be influenced by sampling variations and subjective enumeration. A circulating biomarker that could be measured as part of a routine peripheral blood test would provide a much quicker, safer, and reliable alternative. However, no such biomarker currently exists to predict or monitor MDS/AML disease progression.

ENO1 (also known as α-enolase or phosphopyruvate hydratase) is a ubiquitous glycolytic enzyme with variable expression in response to stress, metabolic and pathologic states of a cell [7]. Despite playing a main role in energy production through anaerobic glycolysis, ENO1 has also been found to shuttle between various cellular compartments and be involved in numerous non-metabolic activities, including cell division, tRNA transport, cytoskeleton organisation, and mitochondrial membrane stability [8]. It is overexpressed in many cancers, and in solid tumours, high cellular ENO1 protein and/or mRNA expression has been linked with poor prognosis for lung [9], breast [10], pancreatic [11], and hepatocellular [12] carcinomas. Gene expression profiling has identified ENO1 as one of the most highly expressed genes in AML [13], and interrogation of The Cancer Genome Atlas (TCGA) repository has correlated high ENO1 mRNA expression with decreased overall survival in AML patients [14]. In addition, studies in vitro have shown that ENO1 can translocate to the extracellular cell membrane and be released into the circulation as a soluble form or on the surface of extracellular vesicles [15]. In this extracellular localisation, ENO1 functions as a plasminogen binding receptor to promote matrix degradation through production of plasmin and activation of collagenases, facilitating migration and invasion of solid tumour cells [7]. In AML, higher circulating levels of ENO1 are significantly correlated with poor risk disease [14]. With the unusual exposure of ENO1 on the surface of tumour cells, patients with solid tumours have been shown to develop autoantibodies to ENO1, the presence of which correlate with diagnosis of osteosarcoma [16] as well as tumours in the lung [9] and liver [17]. Circulating anti-ENO1 antibodies have also been detected in blood from chronic lymphocytic leukaemia patients with progressive disease [18], suggesting a possible predictive role for this biomarker in haematological malignancies. However, no other research has been conducted in this area to date.

Although AML patient samples are known to have high ENO1 gene expression, the biological function of ENO1 in AML is unknown, and the pattern of ENO1 protein expression in the bone marrow of AML patients has not been previously examined. Furthermore, it is unknown if ENO1 is also aberrantly expressed in MDS, or whether autoantibodies to ENO1 exist in patients with either disease. Therefore, the potential utility of ENO1 as a biomarker for MDS and/or AML has not been investigated. The aim of this study was to determine if ENO1 and/or anti-ENO1 autoantibodies were altered in AML and/or MDS and if either correlated with patient prognosis.

## 2. Methods

### 2.1. Meta-Analysis of ENO1 Expression and AML Patient Survival from Online Databases

Expression of ENO1 mRNA associated with AML patient survival was queried in online datasets available from cBioPortal (http://cbioportal.org, accessed on 14 June 2023) [19], PrognoScan (http://dna00.bio.kyutech.ac.jp/PrognoScan/index.html, accessed on 14 June 2023) [20], and PRECOG (http://pregog.stanford.edu, accessed on 14 June 2023) [21]. Nine unique datasets with overall survival (OS) and ENO1 mRNA expression were identified. Datasets with overlapping samples were excluded from the analysis. Patients were divided into low and high expression according to the median ENO1 expression in each study, individual Kaplan–Meier survival curves were generated, and hazard ratios (HR) with 95% confidence intervals (95% CI) were calculated within each online analysis tool. Standard errors (SE) were obtained from the sample size and HR using the equation: SE = HR/√(HRxsample size) [22]. Hazard ratios and standard errors were used to perform meta-analyses by generic inverse variance method using MedCalc v20.123 (MedCalc Software Ltd., Ostend, Belgium). Due to the high heterogeneity (I^2^ = 76.3%, *p* < 0.0001) between the datasets, the HR and 95% CI were calculated using the random effects model that assumes there is a distribution of true effect sizes among the populations studied. Egger’s test was used to ascertain publication bias.

### 2.2. Patient Samples

Bone marrow trephines from 12 MDS patients at diagnosis and 5 AML patients at diagnosis and remission were obtained from the Hunter Cancer Biobank (Hunter Medical Research Institute, New Lambton Heights, NSW, Australia). Peripheral blood plasma was obtained from a separate cohort consisting of 36 AML patients at diagnosis, 11 MDS patients at diagnosis, and 20 previously diagnosed MDS patients who were dependent on red blood cell transfusions. For the latter patients, study bloods were collected immediately prior to transfusions. Normal control plasma was sourced from 20 Australian Red Cross blood donors. This study was approved by the Hunter New England and University of Newcastle Human Research Ethics Committees. All participants provided written informed consent.

### 2.3. Immunohistochemistry

Immunohistochemistry for ENO1 was performed on bone marrow trephines by the Hunter Cancer Biobank using the Ventana Discovery Ultra automated system (Ventana Medical Systems Inc., Tucson, AZ, USA). Briefly, 4 μm sections of formalin-fixed paraffin-embedded tissues were cut onto glass slides and dried at 45 °C for minimum 2 h. Slides were loaded on to the instrument for heat fixing and dewaxing, before blocking and application of anti-ENO1 antibody (1/12,000; Santa Cruz sc10082) for 32 min at 36 °C degrees. A secondary anti-mouse HQ followed by a tertiary HRP HQ were each applied for 20 min at 37 °C. Slides were developed with diaminobenzidine and counterstained with Mayers Haematoxylin before being mounted with glass coverslips. Normal bone marrow was used as control tissue and all products were purchased from Roche unless otherwise specified.

Slides were scanned using an Aperio Scanscope (Leica Biosystems, North Ryde, NSW, Australia) and the number of positively stained cells and the intensity of staining as rated on a scale of 0–300 [H-score = 1  ×  (% of lightly stained cells)  +  2  ×  (% of intermediate stained cells)  +  3  ×  (% of darkly stained cells)] were determined using QuPath [23]. Blast counts were determined from bone marrow aspirates collected at the time of biopsy. Researchers performing the analysis were blinded to patient diagnoses.

### 2.4. Quantification of Circulating Anti-ENO1 Autoantibodies

Plasma anti-ENO1 antibodies were measured by ELISA (MyBioSource MBS2709346, San Diego, CA, USA) according to the manufacturer’s instructions. All tests were performed in duplicate and 5 results with % coefficient of variation >25% were omitted from the analysis, resulting in a total of 82 results with average ± standard deviation %CV = 10.8 ± 6.6% and minimum detection of <1.27 ng/mL. Three MDS patients without clinical data and one outlier were subsequently omitted from the analysis, bringing the total cohort to n = 78. AML patients were classified as having high or low anti-ENO1 levels according to the median value.

### 2.5. Statistical Analysis

All data are presented as median and range unless otherwise specified. Multiple cohorts were compared by Kruskal–Wallis and Jonckheere–Terpstra trend tests, with outliers calculated with Tukey’s method and additional post hoc analysis using Dunn’s test. Correlations between continuous variables were assessed by Spearman rank order correlation coefficient, *r* (*rho*). Probability of survival was illustrated using Kaplan–Meier curves and log-rank test was used to calculate hazard ratios (HR) with 95% CI. Differences in frequencies were assessed by chi-squared test. Statistical analysis was performed using MedCalc Statistical Software v22.016 and *p*-values < 0.05 were considered statistically significant.

## 3. Results

### 3.1. High ENO1 mRNA Expression Is Associated with Worse AML Overall Survival

In order to confirm a previous report correlating ENO1 expression with OS in AML [14], a meta-analysis of ENO1 mRNA expression and AML patient survival was conducted using nine online publicly available datasets, comprising a total of 1419 AML patient samples. Kaplan–Meier survival curves from each individual study indicated that elevated ENO1 expression was significantly associated with reduced patient OS when compared to low ENO1 expression in three of nine (30%) independent datasets (Appendix A), with hazard ratios (HR) > 1 in all nine datasets. Subsequent meta-analysis revealed that high ENO1 expression was significantly associated with worse OS (HR 1.22 [95% CI 1.10–1.34], *p* < 0.001, Figure 1) when all datasets were combined, with no significant publication bias detected (*p* = 0.1941).

### 3.2. ENO1 Protein Is Highly Expressed in AML and MDS Patient Bone Marrow Biopsies at Diagnosis

To identify the level of ENO1 protein expression in AML and MDS patient tumour cells, ENO1 was examined by immunohistochemistry in bone marrow trephine samples obtained from 5 AML patients (2M, 3F; 4 with favourable and 1 with adverse 2022 ELN risk classification [24]) at diagnosis and again at morphological remission, and 12 MDS patients (8M, 4F; disease risk classification unknown) at diagnosis. Staining for ENO1 appeared more intense and widespread in both AML and MDS at diagnosis compared to AML in remission (Figure 2A,C vs. B, respectively). Quantification revealed that both the percentage of positively stained cells and the intensity of staining (H-score) was significantly higher in MDS compared to AML in remission (34.07, 14.78–65.08% vs. 14.99, 10.96–23.95%; *p* = 0.0214 and 54.21, 19.59–107.76 vs. 15.53, 11.98–25.52; *p* = 0.0069, respectively; Figure 2D,E). Most interestingly, however, was the similarity in ENO1 staining between AML and MDS patients at diagnosis, despite the significant difference in the percentage of blast cells between these cohorts (AML: 50.00, 26.20–72.00% vs. MDS: 9.75, 1.00–21.00%; *p* = 0.0353; Figure 2F), suggesting that high ENO1 expression was not restricted to the tumour cells. This was confirmed by a lack of correlation between blast percentages and ENO1 staining (% positive cells *r* = 0.3092, *p* = 0.1607; and H-score *r* = 0.2779, *p* = 0.2104; Figure 2G,H).

### 3.3. Anti-ENO1 Antibodies Are Lowest in AML Patients

We hypothesised that abnormal expression of ENO1 may trigger an autoimmune response that could be used as a biomarker of bone marrow dysfunction prior to the emergence of leukaemic blast cells. We tested this theory by investigating for the presence of anti-ENO1 autoantibodies in a larger cohort of MDS and AML patients and compared the levels to that of normal controls. Circulating anti-ENO1 antibody levels were measured in 34 AML patients at diagnosis, 26 MDS patients, and 18 normal controls (Table 1). Anti-ENO1 antibodies were detected in 63/78 (80.8%) of samples tested, with an unexpected tendency towards patients being more likely than normal controls to have undetectable levels (15/60, 25.0% vs. 1/18, 5.5%; *p* = 0.0750). Overall, anti-ENO1 levels were significantly different between the cohorts (*p* = 0.0024), with AML patients having significantly lower circulating levels of anti-ENO1 autoantibodies compared to controls (median, range = 1.63, 0.00–36.90 vs. 11.24 (0–40.17) ng/mL, *p* = 0.0019, Figure 3A), resulting in a positive test for trend (controls > MDS > AML, *p* = 0.0032). Although there were significant age differences between the groups (*p* = 0.0074), there was no correlation between age and anti-ENO1 autoantibody levels (all subjects combined: *r* = −0.115, 95% CI = −0.329–0.110; *p* = 0.3154, Figure 3B).

MDS patients were further stratified using the revised international prognostic scoring system (IPSS-R) [25] into categories of very low/low risk (n = 12), intermediate risk (n = 5), or high/very high risk (n = 9; Table 2). There were no significant differences in age, sex, or circulating levels of anti-ENO1 antibodies according to disease risk (*p* ≥ 0.05). There were five patients with complex karyotypes (≥3 cytogenetic abnormalities), but there was no significant difference in circulating auto-ENO1 antibody levels between these patients and the rest of the MDS cohort (n = 21) (*p* = 0.4903).

In order to determine if anti-ENO1 autoantibody levels were predictive of worse disease within the AML cohort, AML patients were stratified into favourable (n = 12), intermediate (n = 3), and adverse (n = 19) risk according to the European LeukemiaNet 2022 classification [24]. As indicated in Table 3, there were no significant differences in age, sex, or blast percentages between the groups (*p* ≥ 0.05). Although the adverse risk AML patients recorded the lowest anti-ENO1 concentration (0.920, 0–36.90 ng/mL), this was not significantly different from the intermediate (7.34, 2.75–11.49) or favourable (1.07, 0–34.60) risk groups (*p* = 0.3080). There was also no correlation between anti-ENO1 autoantibody levels and bone marrow blast percentages in AML at diagnosis (*r* = 0.284, *p* = 0.1040; Figure 4A). When patients were stratified by TP53 mutation status, those with a TP53 mutation (n = 7) recorded lower levels of circulating auto-ENO1 antibody levels (median = 0.51, range = 0–4.45 ng/mL) compared to the 27 patients without this poor prognosis factor (1.80, 0–36.90 ng/mL); however, this was not statistically significant (*p* = 0.1381).

In order to determine if anti-ENO1 autoantibody levels were predictive of worse outcomes in AML patients, these were correlated with overall survival in the AML cohort. There were 20 deaths amongst the 34 AML patients, with an overall median survival time of 450 days (95% CI: 173–608 days). Log-rank test for comparison of survival curves indicated no significant difference between patients with low compared to high levels of circulating ENO1 autoantibodies, defined as above or below the calculated median of 1.39 ng/mL (HR = 0.8800, 95% CI: 0.3645–2.1246, *p* = 0.7741, Figure 4B).

## 4. Discussion

ENO1 has been extensively studied in solid tumours, where high cellular expression has been directly correlated with poor clinical outcome, disease progression, and worse overall survival [26]. Although ENO1 is one of the most highly expressed genes in AML [13], its potential role as a biomarker has not been investigated. Here, we used online databases to pool data from 1419 AML patients across multiple studies to demonstrate that high cellular expression of ENO1 is significantly correlated with a 22% worse overall survival. Unfortunately, no similar expression data were available for investigation of ENO1 in MDS. This analysis was further limited by the lack of information to allow for stratification by AML subtypes and treatment regimes, which no doubt contributed to the variation in results observed between individual studies, as reflected by the high heterogeneity (I^2^) score of the meta-analysis. However, all nine studies demonstrated a positive correlation between high ENO1 expression and worse OS.

Mutation/deletions in ENO1 appear to be rare in AML; of the profiled studies, only two [27,28] had mutation and copy number profiling available (CBioportal), showing only 1 homozygous deletion of ENO1 out of 872 samples analysed (http://cbioportal.org, accessed on 14 June 2023) [19]). ENO1 is located on chromosome 1p36.2 in an area harbouring numerous tumour suppressor genes [29,30] and can become a ‘passenger’ deletion in glioblastomas and astrocytic tumours where loss of 1p36 is common [31]. Tumours with ENO1 deletion become reliant on ENO2 for glycolysis and are thus prone to synthetic lethality when ENO2 is pharmacologically targeted [32]. Although loss of heterozygosity of chromosome 1p has been associated with progression of MDS to AML, this has been mapped to involve regions distal to 1p36.3 and therefore less likely to involve ENO1 [33], negating the potential of this therapeutic strategy.

Immunohistochemical staining revealed high expression of ENO1 in BM trephine samples from both AML and MDS patients at diagnosis. This was unexpected, considering that ENO1 overexpression is believed to be tumour associated, and MDS by definition contains significantly lower blast numbers compared to AML. A similar phenomenon has been observed in patients with multiple myeloma, where protein analysis confirmed that ENO1 is not restricted to tumour cells, but also increased in surrounding bone marrow plasmacytoid dendritic cells [34]. Furthermore, we observed a reduction in ENO1 expression in AML remission marrows which are morphologically devoid of AML cells and are considered to have regained a more normal haematopoietic function. These were used as a surrogate for normal bone marrow biopsies, which were unavailable, and displayed similar levels of ENO1 staining as previously reported by normal marrow immunohistochemistry [35]. However, further studies would require a direct comparison to confirm ENO1 protein expression in AML remission compared to normal bone marrow. Taken together, it could be speculated that ENO1 may be a marker of bone marrow dysfunction in these malignancies and could have a potential use as a biomarker for early detection of bone marrow failure preceding numerous hematologic diseases. It would therefore be of particular interest to examine ENO1 expression in bone marrow samples from patients with potential pre-MDS conditions, such as clonal cytopenia of uncertain significance and clonal haematopoiesis with indeterminate potential. These low-grade clonal conditions are increasingly being identified in otherwise healthy older individuals and carry an increased risk of developing haematopoietic neoplasms [36]. They are characterised by MDS-related mutations but lack widespread dysplasia and elevated blast counts [37], thereby maintaining a relatively normal bone marrow function. Examination of ENO1 expression may be useful to be investigated for prognostic impact in these patients.

Both circulating ENO1 and ENO1 autoantibodies have shown promise as ‘liquid biopsies’ for diagnosis and prognosis in solid tumours. Although high serum levels of ENO1 have been previously reported in AML [14], we were unable to confirm this in the present study, which had only plasma samples available for testing. However, the overexpression of tumour associated antigens is often associated with the production of autoantibodies through humeral immune responses [38]. The presence of anti-ENO1 autoantibodies is well documented in patients with solid organ malignancies with varying significance for patient clinical outcomes [39]. However, there are limited data on the presence or significance of ENO1 autoantibodies in haematological malignancies, particularly leukaemias and myelodysplastic neoplasms. As far as we are aware, this is the first such analysis in MDS and AML, and we have demonstrated a trend towards decreasing anti-ENO1 autoantibodies from controls > MDS > AML. Although it could be argued that generally low antibody levels would be an expected reflection of the deleterious effects of disease on the immune system, a study by Shih et al. found no difference in total serum immunoglobulin (Ig) levels (IgA, IgG and IgM) for lung and breast cancer patients with late-stage disease compared to healthy controls. Thus, the observed negative correlation between anti-ENO1 autoantibodies and malignancy was believed to be a specific event and not simply due to a reduction in total immunoglobulin levels in these metastatic solid tumour patients [40]. Unfortunately, this comparative analysis was not performed in the current MDS and AML cohorts; however, previous studies have demonstrated an effective increase in other auto-antibodies in these diseases, suggesting that immunological insufficiency is unlikely to be the cause [41,42]. In contrast, ENO1 autoantibodies were more frequently detected in sera from adult CLL patients with progressive disease than in those with stable disease [18]. In this instance the authors concluded that increased ENO1 autoantibodies represented a clinical inefficacy of the patient sera to trigger complement dependent cytotoxicity and inhibit disease progression.

Most adult studies using comparison cohorts have demonstrated detectable levels of ENO1 autoantibodies in the majority of healthy controls [40,43,44], limiting the use of ENO1 antibodies as a diagnostic marker. However, a single study demonstrating the diagnostic potential of ENO1 autoantibodies in childhood B-ALL showed detectable serum anti-ENO1 in 27% of patients compared to only 4% of controls [45]. The lack of circulating anti-ENO1 in healthy children compared to healthy adults suggests a more promising role for anti-ENO1 in paediatric cancer diagnosis.

Although we did not find a correlation between age and anti-ENO1 antibody levels in our adult cohorts, the control group was significantly younger than the AML and MDS groups, and this may have influenced the finding of decreased anti-ENO1 antibody levels in these patient groups (Figure 3). Ideally, age-matched controls would be employed to eliminate this potential bias. This highlights the main limitation of this study, where the small sample size restricts the conclusions that can be drawn from the analysis. Nevertheless, a better understanding of ENO1 expression in adult AML and MDS may provide a novel way to monitor disease progression in diagnosed patients. In particular, the use of peripheral blood samples for quantifying anti-ENO1 antibody levels serially over time may lessen the frequency of invasive bone marrow biopsy testing and could be used as a complement to tracking genomic markers in blood. Further studies using serial blood sampling on individual patients will be required to determine the clinical usefulness of circulating anti-ENO1 antibody levels.

## 5. Conclusions

This is the first study to investigate bone marrow ENO1 protein expression and circulating anti-ENO1 antibodies in patients with MDS and AML. We found that ENO1 was overexpressed in bone marrow biopsies from both AML and MDS patients, and that this was not restricted to tumour cells. This suggests that high ENO1 expression may be a signal of early bone marrow dysfunction. In addition, circulating autoantibodies to ENO1 were found to be common in normal adults, with a surprising steady decrease in MDS and AML, suggesting a potential use as a biomarker.

## Figures and Tables

**Figure 1 cancers-16-00884-f001:**
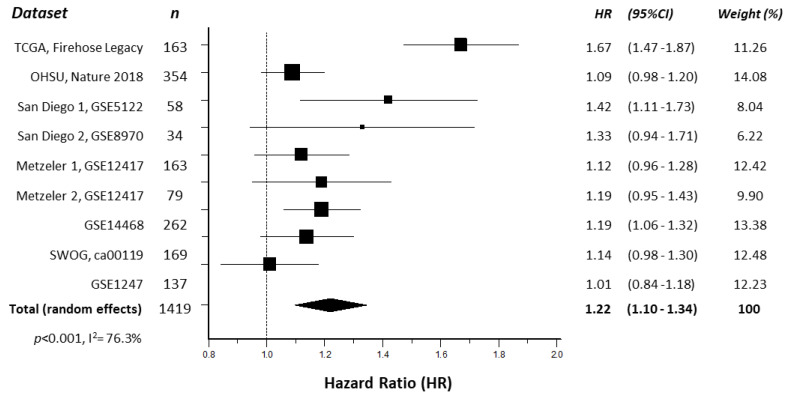
Forest plot illustrating the association of ENO1 expression with AML patient overall survival. The hazard ratios (HR) of individual studies are shown as squares sized to reflect the indicated weight of each study in the overall estimate; 95% confidence intervals (CI) are shown as horizontal lines. The pooled HR from all studies combined (total, random effects) is shown as a diamond. The calculated *p*-value and heterogeneity estimate (I^2^) are indicated.

**Figure 2 cancers-16-00884-f002:**
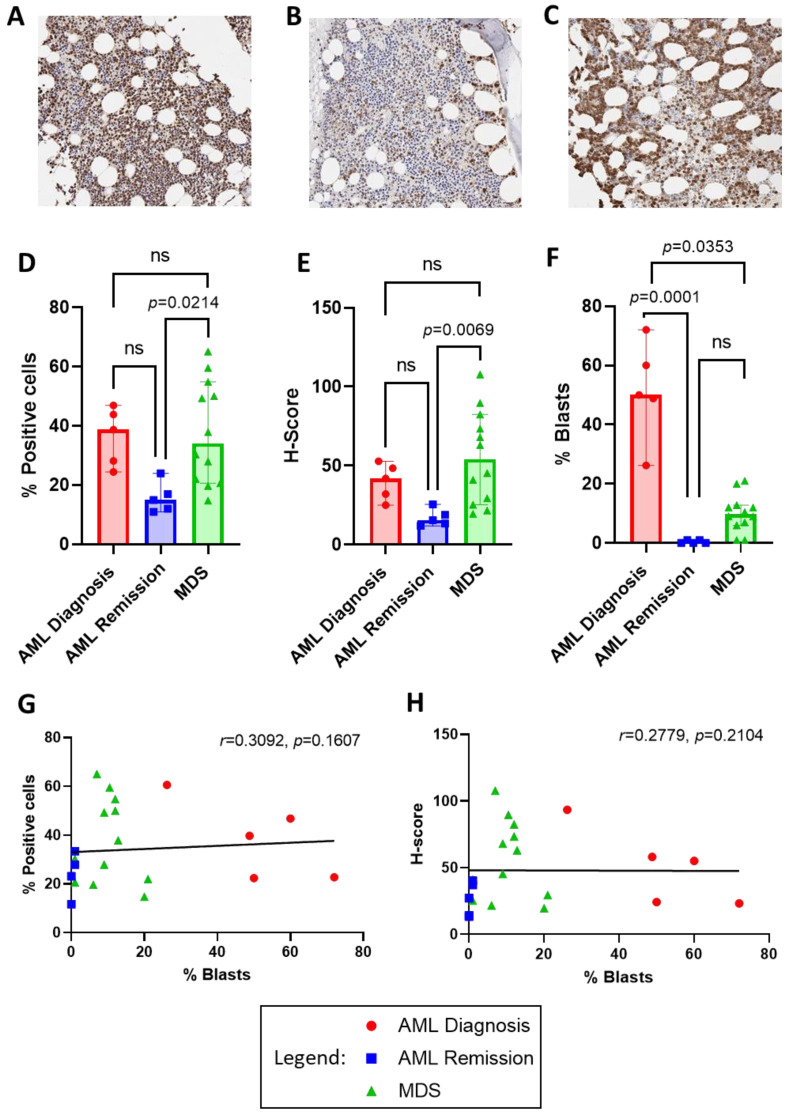
ENO1 protein expression in bone marrow trephine biopsies from AML at diagnosis and remission compared to MDS at diagnosis. Representative low power (20×) micrographs of immunohistochemical staining for ENO1 in (**A**) AML at diagnosis, (**B**) AML at remission, and (**C**) MDS at diagnosis, where presence of ENO1 is indicated by brown staining. QuPath analysis results for the percentage of positive cells (**D**) and H-score staining intensity (**E**) in each cohort. Comparison of blast cell percentages from bone marrow aspirates is illustrated in (**F**), and lack of correlation with ENO1 percentage of positive cell staining in (**G**) and H-score in (**H**). Bar graphs illustrate median/interquartile range. Ns = not significant.

**Figure 3 cancers-16-00884-f003:**
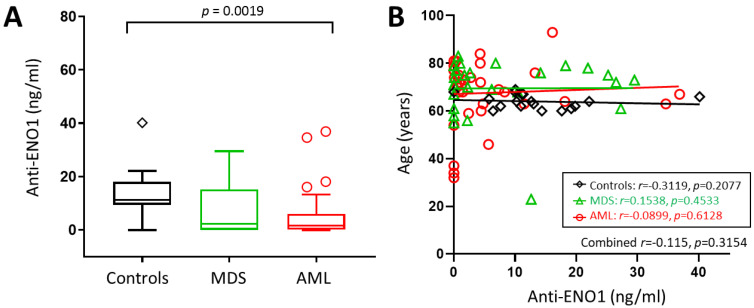
Anti-ENO1 antibodies in AML and MDS patients and controls. (**A**) Box plot showing median anti-ENO1 levels measured for each cohort, with whiskers and outliers calculated using Tukey’s method. (**B**) Scatterplot of age vs. anti-ENO1 levels showing linear trend lines.

**Figure 4 cancers-16-00884-f004:**
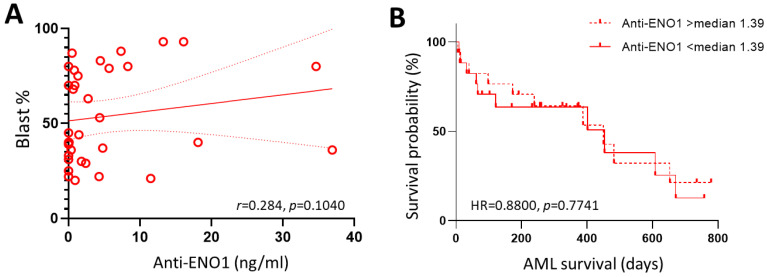
Anti-ENO1 antibodies in AML patients. (**A**) Scatterplot of BM blast percentage vs. anti-ENO1 levels showing linear trend line (solid red line) and 95% confident intervals (dotted red lines). (**B**) Comparison of Kaplan–Meier curves for AML patients with anti-ENO1 levels above and below the calculated median of 1.39 ng/mL.

**Table 1 cancers-16-00884-t001:** Anti-ENO1 antibody levels in AML, MDS, and control cohorts.

Cohort	n	Undetectable Anti-Eno1 N (%)	Anti-ENO1 Median (Range)	Age Median (Range)
Normal controls	18	1 (5.5%)	11.24 (0–40.17)	64 (60–69)
MDS	26	7 (26.9%)	2.24 (0–29.5)	72.5 (23–83)
AML	34	8 (23.5%)	1.63 (0–36.90)	70.5 (32–93)
*p*-value		0.0750 ^a^	0.0024/0.0032 ^b^	0.0074

^a^ normal controls vs. (MDS diagnosis + MDS transfusion dependent + AML), ^b^ Jonckheere–Terpstra trend test.

**Table 2 cancers-16-00884-t002:** Characteristics of the MDS patient cohort.

Variable	MDS IPSS-R Risk Classification			*p*-Value
	Very Low/Low (n = 12)	Intermediate (n = 5)	High/Very High (n = 9)	
Age, years	74.5 (56–83)	71 (55–76)	71 (23–79)	0.5686
Sex, M:F	6:6	3:2	8:1	0.1724
Complex karyotype ^a^/total (%)	0/12 (0%)	0/5 (0%)	5/9 (56%)	0.003
Anti-ENO1, ng/mL	2.43 (0–27.28)	1.73 (0–14.16)	10.43 (0–29.5)	0.6437

Data provided as median (range) unless otherwise specified. ^a^ ≥3 cytogenetic abnormalities.

**Table 3 cancers-16-00884-t003:** Characteristics of the AML patient cohort.

Variable	AML 2022 ELN Risk Classification			*p*-Value
	Favourable (n = 12)	Intermediate (n = 3)	Adverse (n = 19)	
Age, years	73.5 (34–79)	69 (63–74)	68 (32–93)	0.8084
Sex, M:F	10:2	1:3	13:5	0.0942
Blast percentage	72.5 (25–93)	63 (21–88)	40 (20–93)	0.4380
Mutation status, n	NPM1 = 5 CEBPA = 1	NPM1 = 3 FLT3- ITD = 3	TP53 = 7 ASXL1 = 4 RUNX1 = 5	n/a
Anti-ENO1, ng/mL	1.07 (0–34.60)	7.34 (2.75–11.49)	0.920 (0–36.90)	0.3080

Data provided as median (range) unless otherwise specified.

## Data Availability

Online datasets are available from cBioPortal (http://cbioportal.org) [19], PrognoScan (http://dna00.bio.kyutech.ac.jp/PrognoScan/index.html) [20], and PRECOG (http://pregog.stanford.edu) [21]. Patient data is available upon request.

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
