# Peer review of "High Expression of ENO1 and Low Levels of Circulating Anti-ENO1 Autoantibodies in Patients with Myelodysplastic Neoplasms and Acute Myeloid Leukaemia"

_cancers, 2024, doi:10.3390/cancers16050884_

Round 1

Reviewer 1 Report

Comments and Suggestions for Authors

This article provides a very interesting perspective and new data regarding ENO1 in MDS and AML. The authors have done a good job in framing the context, making use of the data stored in the databases, framing the best study cohorts and performing their analyses on them, and identifying how a greater expression of ENO1 is correlated with worse survival. The article goes on to offer data from a new cohort of patients for these diseases studied by the authors, documenting how ENO1 is higher at diagnosis than in the remission phases of AML. They then proceed to look at the antibodies against ENO1, discovering that these levels are lower in AML patients.

Overall, the article offers an important perspective and new data to support the importance of considering ENO1 as a biomarker in these pathologies, with the hope that this may be useful to alleviate the use of invasive and burdensome methods such as bone marrow sampling. Further studies are needed to fully develop this perspective, and despite the good quality of the study, there are some concerns, and their resolution could improve its comprehensibility and clarify some aspects.

In the supplementary figure, I suggest downloading the graphs with the appropriate functions often provided by online tools, rather than publishing entire screenshots, if possible. This would improve the presentation and flow of the article, as it is currently difficult to visually inspect those figures.

In Figure 2G the authors combine two graphs into one. While this is technically possible, I would personally recommend separating them for greater clarity. This is because the 2 scales on the y-axis refer to different things (one in percent and the other a score that can exceed 100). Additionally, when using separate charts, I would suggest documenting the respective statistics and correlation values directly into the chart. This would alleviate the reader's need for interpretation and would immediately make the result clearer.

In the paragraph on line 208 Ant-ENO1 antibodies are lowest in AML and correlate with disease progression from MDS, I generally struggle to understand how this correlates with the disease progression of MDS. Could the authors highlight this here as an answer, or if so clarify further in the article?

Again in the same paragraph, as indicated by the authors in line 222, a significant age difference between the groups was detected which could have influenced the interpretation of the phenomenon. I suggest selecting age-matched control groups to minimize this possible bias, or if this is not possible, including a paragraph in the discussions dedicated to the limitations of this study and mentioning this. In general, I suggest having a paragraph on possible limitations of the study and the research topic.

in Figure 3B it is difficult to understand which groups the authors are referring to. it is conceivable that this could be all the patients in the dataset including the controls. I suggest coding each group with a separate symbol and color, and it is possible to make trend lines for each group and related correlation values and statistics, in addition to the general one. I suggest including these values, in the figure and/or in the legend.

Some of these considerations also apply to Figure 4, where it could be beneficial to report the statistical values.

Regarding the references, I am dubious about references 13 and 14. From a quick check, I found these to be only available in Chinese (except for the abstract). Are there English versions of these articles? In general, regarding all references, I suggest using more recent and high-impact articles if possible.

Author Response

The authors would like to thank the reviewers for their time, insightful comments and overall positive feedback. We have addressed their concerns below and made appropriate changes to the attached manuscript (tracked in MS word). In addition, we found that numbers corresponding to figure 2 were incorrectly entered in the text and we have amended these in the results section on page 5.

Reviewer 1

In the supplementary figure, I suggest downloading the graphs with the appropriate functions often provided by online tools, rather than publishing entire screenshots, if possible. This would improve the presentation and flow of the article, as it is currently difficult to visually inspect those figures.

Author’s response: The screenshots provide much more information than the downloaded graphs, enabling the data to be recreated by any researcher. This is only meant to be a supplementary figure, which will be much larger when not embedded into the final manuscript. We have removed it from the manuscript and broken it up into 3 pages to create a larger, more legible file.

In Figure 2G the authors combine two graphs into one. While this is technically possible, I would personally recommend separating them for greater clarity. This is because the 2 scales on the y-axis refer to different things (one in percent and the other a score that can exceed 100). Additionally, when using separate charts, I would suggest documenting the respective statistics and correlation values directly into the chart. This would alleviate the reader's need for interpretation and would immediately make the result clearer.

Author’s response: As suggested, we have divided figure 2G into two separate graphs (2G and 2H) and included statistical values on all charts instead of in the legends. We have adjusted the colours to be consistent throughout the figures.

In the paragraph on line 208 Ant-ENO1 antibodies are lowest in AML and correlate with disease progression from MDS, I generally struggle to understand how this correlates with the disease progression of MDS. Could the authors highlight this here as an answer, or if so clarify further in the article?

Author’s response: We acknowledge that the term ‘progression’ is misleading and we have removed it from this line, the introduction, the conclusion, and the discussion (originally located at line 319) and replaced it in the discussion with:

………….We have demonstrated a trend towards decreasing anti-ENO1 autoantibodies from controls>MDS>AML.

Again in the same paragraph, as indicated by the authors in line 222, a significant age difference between the groups was detected which could have influenced the interpretation of the phenomenon. I suggest selecting age-matched control groups to minimize this possible bias, or if this is not possible, including a paragraph in the discussions dedicated to the limitations of this study and mentioning this. In general, I suggest having a paragraph on possible limitations of the study and the research topic.

Author’s response: As suggested, we have added a paragraph at the end of the discussion discussing the limitations of the study:

Although we did not find a correlation between age and anti-ENO1 antibody levels in our adult cohorts, the control group was significantly younger than the AML and MDS groups, and this may have influenced the finding of decreased anti-ENO1 antibody levels in these patient groups (Figure 3). Ideally, age matched controls would be employed to eliminate this potential bias. This highlights the main limitation of this study, where the small sample size restricts the conclusions that can be drawn from the analysis.

in Figure 3B it is difficult to understand which groups the authors are referring to. it is conceivable that this could be all the patients in the dataset including the controls. I suggest coding each group with a separate symbol and color, and it is possible to make trend lines for each group and related correlation values and statistics, in addition to the general one. I suggest including these values, in the figure and/or in the legend.

Author’s response: As suggested, we have colour coded the groups with separate symbols and added individual trend lines and statistics in the figure.

Some of these considerations also apply to Figure 4, where it could be beneficial to report the statistical values.

Author’s response: As suggested, we have included the statistical values and recoloured the graphs in red to maintain consistency with other figures colour scheme.

Regarding the references, I am dubious about references 13 and 14. From a quick check, I found these to be only available in Chinese (except for the abstract). Are there English versions of these articles? In general, regarding all references, I suggest using more recent and high-impact articles if possible.

Author’s response: We have removed these references and added more recent references where possible.

Reviewer 2 Report

Comments and Suggestions for Authors

It was a pleasure to review this interesting original study by Lincz et al on the potential use of ENO1 levels and antiENO1 circulating antibodies as a diagnostic and prognostic tool in patients with AML and MDS. The authors have made a significant effort to approach the subject, based on data on solid tumors and using data from online databases as well as a rather small number of AML and MDS bone marrow and blood samples.

Please find my comments below.

·       One of the main drawbacks of the study is the rather small number of patient samples studied. I understand that this cannot be improved at this stage of the study, but the authors should add a comment in the discussion in a section on the strengths and limitations of their study.

·       The term “myelodysplastic syndrome” should be replaced by “myelodysplastic neoplasm” as per the WHO 2022 classification.

·       Lines 46-49: The definition provided here for MDS is not very accurate. Please revise.

·       Line 54: MDS and AML are certainly not accepted as a continuum of the same disease. Please revise. This consideration also affects the discussion, where differences between AML and MDS in the expression of ENO1 are considered unexpected.

·       Is there a biological role of ENO1 in general  (in health) and most importantly in AML/MDS? If so, the authors should provide some information on that. I would prefer this information to be added in the introduction rather than the discussion. It would help the reader understand the rationale of the study.

·       What is the rationale for comparing the intensity of staining for ENO1 between patients with MDS and patients with AML in remission? These two settings are totally different.

·       Line 232: normal karyotype is not a significant parameter for the prognosis of patients with MDS per se.  Instead, complex karyotype, or other karyotypic changes with prognostic significance should be reported here (a difference between the IPSS-R risk groups should be then noted). Moreover, other prognostic significant parameters (blast count, cytopenias) should be reported here.

·       Line 235: The term “abnormal cytogenetics” is of no prognostic significance (for example both del(5q) and -7 are abnormal, but with vast differences in their prognostic significance). A cytogenetic risk stratification should be used instead here. The same applies to comparisons in AML patients (lines 248-251).

·       There is a problem in the columns of Table 2. Please fix it.

·       Lines 296-301. This assumption is rather arbitrary.

·       In order for a test to be used as a diagnostic test for a condition (in this case MDS), there should be at least a comparison of ENO1 expression levels between patients with cytopenias/dysplasia of other causes and those with MDS. Thus, proposing ENO1 as a diagnostic test for MDS is at least an exaggeration, based on the available evidence.

Author Response

The authors would like to thank the reviewers for their time, insightful comments and overall positive feedback. We have addressed their concerns below and made appropriate changes to the manuscript (tracked in MS word). In addition, we found that numbers corresponding to figure 2 were incorrectly entered in the text and we have amended these in the results section on page 5.

Reviewer 2

  • One of the main drawbacks of the study is the rather small number of patient samples studied. I understand that this cannot be improved at this stage of the study, but the authors should add a comment in the discussion in a section on the strengths and limitations of their study.

Author’s response: As suggested, we have added a paragraph at the end of the discussion discussing the limitations of the study:

Although we did not find a correlation between age and anti-ENO1 antibody levels in our adult cohorts, the control group was significantly younger than the AML and MDS groups, and this may have influenced the finding of decreased anti-ENO1 antibody levels in these patient groups (Figure 3). Ideally, age matched controls would be employed to eliminate this potential bias. This highlights the main limitation of this study, where the small sample size restricts the conclusions that can be drawn from the analysis.

  • The term “myelodysplastic syndrome” should be replaced by “myelodysplastic neoplasm” as per the WHO 2022 classification.

Author’s response: we have changed the wording in the title and throughout the manuscript as suggested (6 changes in total)

  • Lines 46-49: The definition provided here for MDS is not very accurate. Please revise.

Author’s response: We have changed this to:

…..myelodysplastic neoplasms (MDS), a heterogeneous group of clonal myeloid disorders characterised by cytopaenia and dysplastic changes in bone marrow haematopoeitic cells [2]. These diseases typically affect the elderly and carry a high risk of transformation to AML [3]

  • Line 54: MDS and AML are certainly not accepted as a continuum of the same disease. Please revise. This consideration also affects the discussion, where differences between AML and MDS in the expression of ENO1 are considered unexpected.

Author’s response: As suggested, we have revised this sentence to:…..While MDS, MDS/AML, and AML share many genomic abnormalities…..

However, the statement in the discussion is correct, as we did not expect ENO1 expression to be as high in MDS as in AML due to the reduced number of blast cells in MDS.

  • Is there a biological role of ENO1 in general  (in health) and most importantly in AML/MDS? If so, the authors should provide some information on that. I would prefer this information to be added in the introduction rather than the discussion. It would help the reader understand the rationale of the study.

Author’s response: As suggested, we have added the following information to the introduction:

ENO1 (also known as a-enolase or phosphopyruvate hydratase) is a ubiquitous glycolytic enzyme with variable expression in response to stress, metabolic and pathologic states of a cell [7]. Despite playing a main role in energy production through anaerobic glycolysis, ENO1 has also been found to shuttle between various cellular compartments and be involved in numerous non-metabolic activities, including cell division, tRNA transport, cytoskeleton organisation, and mitochondrial membrane stability[8]………………………… In this extracellular localisation, ENO1 functions as a plasminogen binding receptor to promote matrix degradation through production of plasmin and activation of collagenases, facilitating migration and invasion of solid tumour cells[7]…………………. Although AML patient samples are known to have high ENO1 gene expression, the biological function of ENO1 in AML is unknown,

  • What is the rationale for comparing the intensity of staining for ENO1 between patients with MDS and patients with AML in remission? These two settings are totally different.

Author’s response: Unfortunately we did not have access to comparable normal bone marrow tissue sections that could be used as controls for the patient samples included in this study. We therefore opted for AML in remission, containing no blast cells, as a surrogate.  The 14.99% staining observed (Figure 2D) is comparable to that reported for normal bone marrow immunohistochemistry (n=11) by Chung et al.. We have added the following explanation to paragraph 4 of the discussion:

….. we observed a reduction of ENO1 expression in AML remission marrows which are morphologically devoid of AML cells and are considered to have regained a more normal haematopoietic function. These were used as a surrogate for normal bone marrow biopsies, which were unavailable, and displayed similar levels of ENO1 staining as previously reported by normal marrow immunohistochemistry [35]. However, further studies would require a direct comparison to confirm ENO1 protein expression in AML remission compared to normal bone marrow.

  • Line 232: normal karyotype is not a significant parameter for the prognosis of patients with MDS per se.  Instead, complex karyotype, or other karyotypic changes with prognostic significance should be reported here (a difference between the IPSS-R risk groups should be then noted). Moreover, other prognostic significant parameters (blast count, cytopenias) should be reported here.

Author’s response: We have replaced ‘normal karyotype’ with ‘complex karyotype’ in table 2 and reported the results in the text above it. Because the IPSS-R stratification already incorporates all relevant prognostic parameters of blast count, cytopenias, and cytogenetic scoring, we feel this is the best way to classify the small sample size.

  • Line 235: The term “abnormal cytogenetics” is of no prognostic significance (for example both del(5q) and -7 are abnormal, but with vast differences in their prognostic significance). A cytogenetic risk stratification should be used instead here. The same applies to comparisons in AML patients (lines 248-251).

Author’s response: We have removed ‘normal karyotype’ from table 3 and instead reported the ENO1 results associated with TP53 mutation as a matter of interest. Because the ELN Risk classification already incorporates all relevant prognostic parameters, we feel this is the best way to classify the cohort. Due to the small sample size, further stratification does not add any additional meaningful information.

 There is a problem in the columns of Table 2. Please fix it.

Author’s response: These were formatted by the journal after submission. We have corrected the formatting in all tables.

  • Lines 296-301. This assumption is rather arbitrary.

Author’s response: We agree that this sentence is highly speculative, and we have changed the wording to acknowledge this:

Taken together, it could be speculated that ENO1 may be a marker of bone marrow dysfunction in these malignancies, and could have a potential use as a biomarker for early detection of bone marrow failure preceding numerous hematologic diseases.

  • In order for a test to be used as a diagnostic test for a condition (in this case MDS), there should be at least a comparison of ENO1 expression levels between patients with cytopenias/dysplasia of other causes and those with MDS. Thus, proposing ENO1 as a diagnostic test for MDS is at least an exaggeration, based on the available evidence.

Author’s response: We have removed this sentence from the conclusion. We have also removed references to a diagnostic test from Simple Summary and Abstract, and replaced it with:

Simple Summary………. This study examines the potential use of α-enolase (ENO1) as a potential biomarker and shows that it is highly expressed in bone marrow biopsies from patients with both MDS and AML, and that patients with these diseases also have decreased circulating levels of auto-antibodies to ENO1.

Abstract………. BM immunostaining for ENO1 and patient monitoring of anti-ENO1 autoantibody levels may be useful biomarkers for MDS and AML.